# Brazilian Coffee Production and the Future Microbiome and Mycotoxin Profile Considering the Climate Change Scenario

**DOI:** 10.3390/microorganisms9040858

**Published:** 2021-04-16

**Authors:** Deiziane Gomes dos Santos, Caroline Corrêa de Souza Coelho, Anna Beatriz Robottom Ferreira, Otniel Freitas-Silva

**Affiliations:** 1Food and Nutrition Graduate Program, Federal University of State of Rio de Janeiro, UNIRIO. Av. Pasteur, 296, Rio de Janeiro 22290-240, Brazil; deizigomes08@gmail.com (D.G.d.S.); caroline.coelho@unirio.br (C.C.d.S.C.); 2Embrapa Food Agroindustry, Brazilian Agricultural Research Corporation, Rio de Janeiro 23020-470, Brazil; anna.ferreira@embrapa.br

**Keywords:** coffee, climate, global warming, mycotoxins

## Abstract

Brazil holds a series of favorable climatic conditions for agricultural production including the hours and intensity of sunlight, the availability of agricultural land and water resources, as well as diverse climates, soils and biomes. Amidst such diversity, Brazilian coffee producers have obtained various standards of qualities and aromas, between the arabica and robusta species, which each present a wide variety of lineages. However, temperatures in coffee producing municipalities in Brazil have increased by about 0.25 °C per decade and annual precipitation has decreased. Therefore, the agricultural sector may face serious challenges in the upcoming decades due to crop sensitivity to water shortages and thermal stress. Furthermore, higher temperatures may reduce the quality of the culture and increase pressure from pests and diseases, reducing worldwide agricultural production. The impacts of climate change directly affect the coffee microbiota. Within the climate change scenario, aflatoxins, which are more toxic than OTA, may become dominant, promoting greater food insecurity surrounding coffee production. Thus, closer attention on the part of authorities is fundamental to stimulate replacement of areas that are apt for coffee production, in line with changes in climate zoning, in order to avoid scarcity of coffee in the world market.

## 1. Introduction

Brazil is the fifth largest country in geographical area and the largest in cultivatable land (851 million acres) and possesses a series of favorable climatic conditions for agricultural production. Among natural factors that contribute to agricultural production are hours and intensity of sunlight, availability of arable land and water resources, as well as diversity of climates, soils and biomes that favor production of different cultures. Climate diversity is composed of equatorial, tropical, subtropical, temperate, semiarid regions, which places Brazil amongst the largest producers and exporters of food, including coffee, sugar and orange juice, worldwide [1].

Coffee is native to Africa, while arabica coffee is supposedly originated in Ethiopia and robusta from the Atlantic coast (Kouilou region around Angola) and from the African Great Lakes. Most of the world’s coffee is produced in Latin America, particularly in Brazil, which has dominated the world’s coffee production since 1840 [2]. 

Coffee was brought to Brazil from French Guiana by Sergeant Major Francisco e Mello Palheta through the city of Belém in the state of Pará in 1727. Due to favorable edaphoclimatic conditions, coffee culture spread from northern Brazil to various states. Favorable climatic circumstances, soil, and topography solidified the culture in the Paraiba River Valley (Vale do Rio Paraíba) in the states of Rio de Janeiro and São Paulo in the mid-19th century, giving way to a new economic cycle in Brazil. With the introduction of coffee in the international market, it quickly became the main exportation product in Brazil. In 1850, Brazil was already the largest producer, responsible for 40% of the world’s coffee production [3].

According to the Brazilian Ministry of Agriculture, Livestock and Food Supply [3], Brazil has favorable climatic conditions for producing coffee in 15 regions. This diversity guarantees a variety of coffees from the north to south of the country. Amidst such diverse climates, altitudes and types of soil, Brazilian coffee producers have obtained various standards of qualities and aromas, between arabica and robusta species, which each present a wide variety of lineages. Arabica coffee (*Coffea arabica* L.) provides the consumer with a finer, exquisite and better-quality product. This type of coffee is cultivated in altitudes over 800 m from sea level, thus being more predominant in the states of Minas Gerais, São Paulo, Paraná, Bahia, Rio de Janeiro and parts of Espírito Santo. On the other hand, robusta, or conilon (*Coffea canephora*), is mostly used for production of soluble coffees and in certain mixtures with arabica. Robusta presents a characteristic flavor, lower acidity and higher caffeine content, being produced in the states of Espírito Santo, Rondônia and in parts of Bahia and Minas Gerais [3]. 

*C. arabica* normally grows in areas with average temperatures of 24 to 26 °C, however, prolonged exposure to 23 °C or more may result in accelerated flowering and loss of quality. Exposure to temperatures above 30 °C may lead to abnormalities and severe atrophy [4,5]. The differences in temperature tolerance are related to the distribution of precipitation throughout the year and the average soil humidity. Annual precipitation of 1000 mm is considered minimal for the cultivation of *C. arabica*, although some varieties have been known to grow below 762 mm [5]. 

According to Koh et al. [6], temperatures in coffee producing municipalities have increased approximately 0.25 °C per decade since 1974. Annual precipitation has been declining during flowering and maturation, however, since 2002, precipitation during harvest has increased. The municipalities in the northern region of Bahia, north of Goiás and Minas Gerais present the largest average temperatures. Since 2010, average temperatures in these regions frequently exceed the ideal range for arabica coffee (< 23 °C). During the flowering period, temperatures in all states have increased over 1.2 °C. In Bahia, Minas Gerais and São Paulo, these large increases in temperatures have been associated to reduction in rains (>10% reduction). Excessively high temperatures associated to lack of rains may inhibit both the initial budding of flowers as well as the maturation of beans. 

According to Ramirez-Villegas et al. [7], the agricultural sector may face serious challenges in the upcoming decades due to crop sensitivity to water shortages and thermal stress. Furthermore, the increase in temperatures may reduce the quality of cultures and increase pressure by pests and diseases, reducing agricultural production worldwide [8].

Agriculture is expected to play an important role in the context of climate change, not only because it is considered one of the most vulnerable sectors, but also because it is part of the solution since it has the potential to ease greenhouse gas emissions [9,10]. Climatic variability has always been the main factor responsible for fluctuations in coffee productivity worldwide. Thus, climate change, as a result of global warming, is expected to lead to major shifts in where and how coffee will be produced in the future [11]. 

International markets have a strong impact on the Brazilian economy since commodity exports are often restricted by international commercial policies based on food safety. Reducing rejection of foods by the European Union and other countries has become one of Brazil’s national priorities, from both an economic and food safety standpoint [12].

## 2. Vulnerability of Coffee Production to Climate Change

The Intergovernmental Panel on Climate Change (IPCC, 2001) defined vulnerability to climate change as the degree by which a system is susceptible or incapable of dealing with the adverse effects of climate change, including climatic extremes and variability. Furthermore, vulnerability depends on the nature, magnitude and rate of climate change, as well as the variation that a system is exposed to, its sensitivity and adaptability. Exposure refers to the nature and extension of changes that the climate of a certain location is subjected to regarding variables such as temperature, precipitation and extreme climatic events. Sensitivity is a measure of how systems may be affected by the change in climate (for example, how a crop is affected or how human health is impacted). In contrast, adaptability is defined as the capacity of a system to adjust to climate change in order to mitigate possible damage.

According to Baca et al. [13], vulnerability in the livelihoods of small coffee producers is due to three factors: exposure, sensitivity and adaptability. These factors are associated to the interaction between climate change and access to and availability of resources for family farmers. Exposure is quantified by a model of coffee crop suitability comparing current and future climates, representing how the means of subsistence of farming families will be affected by changes in climate. Sensitivity and adaptability are measured by indicators based on family resources such as human, natural, social, physical and financial capital.

In a development project that sought to facilitate adaptation to climate change among coffee producers in Mexico, Nicaragua, Guatemala and El Salvador, farms located in highly vulnerable areas will not have adequate conditions to produce quality coffee until 2050. These conditions include changes in climatic events such as temperature and precipitation, high variability in coffee production, high migratory levels in certain communities, low adaptability in post-harvest infrastructure and in Guatemala and Mexico, low access to credit. In areas that continue to be adequate for coffee growing, albeit with some reduction in adequacy, better agronomic management could reduce the impact of climate change, while in areas in which a low aptitude for coffee production was foreseen, farmers will have to identify alternative crops [13]. 

Fain et al. [14] evaluated vulnerability and future climate adequacy for the cultivation of coffee on the Caribbean island of Puerto Rico. According to projections, they report that warming tendencies may surpass important temperature limits over the next decades. They also point out that warming tendencies may even accelerate after 2040, and with that, result in losses of 60-84% in cultivation conditions until 2070 in what are now considered high yield municipalities. High temperatures and low levels of precipitation may result in lower quality and productivity, aside from greater exposure and sensitivity to certain insects and diseases. The period from 2040 to 2070 may represent an inflection point at which average temperatures on the entire island may exceed optimum parameters for cultivation of *C. arabica*. 

Jaramillo et al. [15] reported that the negative effects of climate change are already evident for many coffee growers in the tropics and for industries. The coffee berry borer (*Hypothenemus hampei*), which is considered a coffee pest worldwide, has already benefited from the rise in temperature in East Africa, where major damage to coffee crops and expansion in its distribution area has been reported. The situation with *H. hampei* is expected to worsen in the current production areas of *C. arabica* in Ethiopia, in the Ugandan part of Lake Victoria and Mt. Elgon regions, Mt. Kenya and the Kenyan side of Mt. Elgon and most of Rwanda and Burundi. The hypothetical estimated number of generations per year of *H. hampei* is expected to increase in all areas producing *C. arabica* from five to ten, thus causing serious implications for the production of *C. arabica* in East Africa.

A study of East African Kihansi coffee, restricted to the Kihansi Gorge in Tanzania, provided an example of how coffee is influenced by pests under accelerated climate change. This local change coincided with a parasitic infestation that undermined the potential of this coffee, with negative consequences for the survival of coffee species in that region [16].

Koh et al. [6] evaluated sensitivity of coffee productivity towards variation in temperature and precipitation from 1974 to 2017 in certain regions in Brazil, in order to map climatic risks to coffee and generate a vulnerability index to identify regions theoretically less capable of adapting to these risks. The authors observed an increase in temperatures in coffee producing municipalities in Brazil and a decrease in annual precipitation during flowering and maturation periods from 1974 onwards. This historic climatic change resulted in 20% reduction in coffee production in southeast Brazil. Minas Gerais, the largest coffee producing state in Brazil, presents one of the largest climate threats and risks in general, which is aggravated by continued expansion of coffee. In the municipalities in the north of Minas Gerais and Rio de Janeiro, greater vulnerability was observed because of smaller coffee harvest, less knowledge, social capital and access to technical assistance, as well as deficient transportation infrastructure. They also point out that Minas Gerais and Rio de Janeiro have the highest average climatic risk in all classification systems due to the combination of high exposure, threats and vulnerability. The states of Paraná and Goiás presented the lowest average risk. Finally, they highlighted that the historical change in climate has already had a substantial negative impact on productivity. This climatic threat, however, is not distributed uniformly across the country, and is mostly concentrated in southeast Brazil, including Minas Gerais, the region with the largest production of arabica coffee. 

According to a study carried out by Tavares et al. [17], a large part of southeast Brazil may suffer significant changes in areas that are currently suitable for growing arabica coffee. In 2018, most of southeast Brazil was suitable for coffee cultivation, varying from totally suitable to at most regular, due to thermal excess or deficiency. However, throughout the XXI century, projections showed strong reduction in completely suitable areas, an increase in regular areas and the rise of inadequate areas. These area restrictions are caused by a 4 to 8 °C average increase in air temperature. The state of Minas Gerais is projected to see a potential 25% reduction in arabica coffee productivity by the end of the XXI century. In this region, suitable areas may be relocated to regions of higher altitude, which in turn results in more challenging farm management, mainly due to operating agricultural machinery in mountainous zones. Thus, to maintain the qualitative and quantitative production of coffee in southeast Brazil, adaptation measures that take into account at least a 2 to 4 °C temperature increase and consider the genetic and physiological traits of arabica coffee cultivars in the region, need to be proposed.

Better coffee varieties and agricultural loans for irrigation and agroforestry systems may provide coffee farmers a way to maintain or improve their productivity while facing climatic threats. In the meantime, development of infrastructure and human capacity in existing cooperatives may help increase access to added value marketing opportunities to compensate for lower revenues [6].

## 3. Potential Climate Change Mitigation Strategies 

Battilani et al. [18] reported that changes in mycotoxin occurrence patterns, such as aflatoxins in crops in Europe due to climate change, are a cause for concern. The authors pointed that official control measures contribute to the global effort to reduce the risks of exposure to aflatoxins through food and feed intake by humans and animals, however, specific action plans need to be directed to the production chain. In addition to these measures, other recommendations have been suggested to minimize the impacts of global climate change such as (i) the use of the modeling approach as a support tool to strengthen the management of aflatoxin to prevent human and animal exposure, (ii) the adoption of new strategies supported by forecasts, (iii) biological control through the use of atoxigenic strains capable of displacing the toxigenic populations of the fungus (iv) use of risk maps as a communication tool for stakeholders, especially for farmers and livestock keepers, (v) management tools to highlight areas at risk of mycotoxins contamination, in order to prioritize their control and intervention strategies [19,20,21].

For coffee producers various adaptation and mitigation strategies have been proposed in response to challenges faced by the sector, according to International Coffee Organization (ICO) [11]. Short-term adaptation strategies include better agricultural practices and post-harvest processing. Long-term strategies include capacity building, enhanced monitoring of climatic data, increased soil fertility, introduction or preservation of different production models and development of drought and disease resistant varieties. In more extreme cases, a solution may be to diversify or transfer production to more adequate areas. 

Camargo [22] describes some agronomical techniques that may be used to mitigate extreme meteorological events and face the challenges of coffee cultivation such as climate variability or global warming. Some techniques, such as the use of shading (afforestation), high density planting, vegetated soil, optimized irrigation, and agronomical adaptation focusing on crop improvement programs may be used to attenuate the impact of unfavorable temperatures on coffee production. Table 1 presents a summary of studies conducted on coffee production in the wake of global warming, including a description of the effect of possible scenarios on coffee production as well as mitigation strategies.

## 4. Mycotoxins in Coffee in Brazil 

As with other agricultural products, coffee cherries may be contaminated and colonized by a wide variety of microorganisms, including toxigenic species, in the pre- and post-harvest periods. Microbiological studies carried out with green and processed coffee beans have reported that fungi, such as *Aspergillus*, *Fusarium* and *Penicillium*, are natural contaminants of coffee, and may occur during cultivation, processing, transportation, and storage. If environmental conditions are favorable, such as high temperature and humidity, some of these fungi may produce mycotoxins. 

The most frequently reported mycotoxin in coffee is ochratoxin A (OTA), occurring at variable levels, while references to aflatoxins and other toxins are less common [27,28,29].

Microorganisms are a natural part of the plant, thus, in a healthy plant, there is a balance between these commensal organisms and the plant itself. While serious fungal pathogens do afflict coffee, generally, OTA producing fungi are not those responsible for plant diseases. Many are involved in fruit deterioration and may also grow and survive on viable and healthy seeds. Robusta coffee is more resistant to disease than arabica and although the chromosome number of these two species differs, breeding is possible to improve disease resistance in arabica [30].

According to FAO [30], OTA is a heat stable fungal metabolite produced by a proportion of isolates of certain species of the genre *Aspergillus* and *Penicillium*. Few species of *Aspergillus* are known to produce OTA in coffee, namely *Aspergillus ochraceus*, *A. westerdijikiae*, *A. carbonarius* and *A. niger*. The toxin is produced by a micelle within certain levels of physical water activity (aw), nutrition and temperature, all of which are potential control points. In order for it to be produced, OTA producing fungi should remain in favorable conditions for enough time. An essential condition is the availability of water, where an aw > 0.95 is considered too humid and ideal for growth of hydrophilic fungi, including yeast, who will prosper and limit the amount of OTA producing fungi. An aw < 0.80 is considered too dry and OTA producing fungi are unable to produce toxins. An aw between 0.76 and 0.78 is unsuitable for these fungi to grow. Thus the importance of controlling the time in which coffee beans are laid out to dry and reducing the availability of water to avoid the growth of OTA producing fungi [12,30,31].

The various climates and production systems confer different risks to the development of OTA producing fungi. In shaded coffee plantations, the soil maintains its humidity, even in the dry season. In some regions, the harvest generally extends for three months and coincides with a rainy season or humid conditions. In these scenarios, there is a high chance that fallen beans become highly contaminated by fungi. In non-shaded production systems, where the harvest is carried out during the dry season, the risk of contamination is reduced [32]. 

In the early 1900s, the European Union (EU) program for regulating food contaminants, including mycotoxins, pointed out the need to examine the contamination of food by mycotoxins at the global level. Tackling rejection of food by the EU and other countries has become a national priority, from an economic and food security standpoint [12].

Based on the risk that OTA represents to human health, the EU developed regulation for maximum limits of the toxin in a variety of products including coffee. Maximum levels of OTA in coffee are 5 μg/kg for roasted and ground coffee beans and 10 μg/kg for instant coffee (European Commission (EC), 2006). In 2021, the Brazilian Health Regulatory Agency (ANVISA) revised regulation for six mycotoxins (aflatoxins, ochratoxin A, fumonisins, zearalenone, deoxynivalenol and patulin) in over 20 categories of foods. In roasted and soluble coffee the established limits for OTA were 10 μg/kg [33]. 

According to Paterson & Lima [34], ochratoxigenic fungal contamination in coffee beans is affected by many factors such as climate, plant susceptibility, intrinsic and extrinsic factors, product cultivation, handling and substrate nutrients. However, temperature is the abiotic factor that most influences fungal physiology, which may or may not favor growth and toxin production. OTA has nephrotoxic, carcinogenic, teratogenic and immunosuppressing properties and it is the main fungal contaminant of coffee beans cultivated in Brazil, where they are mostly produced by *A. ochraceus*.

Batista et al. [35] analyzed the incidence of ochratoxin A in coffee samples (*Coffea arabica* L.) obtained from farms in 10 southern municipalities of Minas Gerais, and observed greater incidence of filamentous fungi in swept coffee, on the ground and in floating coffee. Authors found *Aspergillus ochraceus*, *Aspergillus sulphureus* and *Aspergillus sclerotiorum* species to be producers of ochratoxin A. OTA was not detected in 44% of samples, while in 31%, it was present between 0.1 and 5.0 μg/kg. Another 25% presented above 5.0 μg/kg of contamination. The study showed that harvest and pre-processing operations impact the characteristics of the coffee and lead to different risks of exposure to contamination by toxigenic fungi and OTA. The greatest risk of exposure was due to contact of the fruit to the soil, mainly constituted by the fraction of coffee that was swept from the ground and by inadequate post-harvesting management during drying on the ground. They highlight that patios on the ground should be avoided, since soil is the natural habitat of ochratoxigenic fungi as well as other microorganisms, and that the adoption of good agricultural practices will lead to a significant reduction in risk of contamination by microorganisms as well as a reduction in OTA.

Coffee producing regions in the Cerrado biome of Minas Gerais and Sorocabana in the state of São Paulo were studied to evaluate the presence of ochratoxigenic fungi and OTA in defective coffee beans. In the study, *Aspergillus westerdijkiae* and *Aspergillus nigri* were encountered in both regions while *Aspergillus carbonarius* was only found in Cerrado, MG. Sour and black beans presented the highest concentration of OTA of 11.3 μg/kg and 25.7 μg/kg, respectively. Although green defective immature beans presented the highest proportion (38%), the presence of ochratoxigenic fungi and OTA was low [36].

Sousa et al. [37] analyzed the species distribution of toxigenic fungi in coffee cultivating regions of the state of Minas Gerais and found a statistical difference between the regions with respect to *Aspergillus nigri* and *Aspergillus circumdatti* incidence. The Zona da Mata region, which is characterized by relatively high temperature and humidity in coffee cultivation and drying regions presented the greatest incidence of these species, followed by the Cerrado and the south of Minas Gerais. They reported that 75% of fungi isolated from the Zona da Mata region belonged to *A. nigri* and 20% to *A. circumdatti*. Furthermore, 41% of the species found in the Cerrado region belonged to *A. nigri* and 11% to *A. circumdatti*. In the south of Minas Gerais state, 19% of the isolated fungi belonged to *A. nigri* and only 1% to *A. circumdatti*. Various factors are responsible for the occurrence of potential OTA producing fungal species in coffee, including environmental conditions, such as humidity and temperature, as well as intrinsic factors, such as water activity [37].

Taniwaki et al. [12] investigated the relationship between the production of OTA on coffee, local climatic conditions and processing factors in different cases and observed that the occurrence of OTA was sporadic and limited to post-harvest. In the first case, in the southeast of the state of São Paulo, a relatively cold and rainy region (average temperature of 18 °C and 66 mm/month precipitation), of low altitude (<800 m), that generally produces low quality coffee, a sample segregated as “floater” during storage was contaminated with 110 μg/kg of OTA. Authors report that a series of factors may have contributed to the contamination: the farm was located in a valley which was often affected by strong fog, during the drying process coffee was often spread in a thick layer and rarely overturned, the barn showed signs of humidity and birds had access to the storage barn, all of which favored the growth of *A. ochraceus* and production of OTA. *A. ochraceus* was isolated in 30% of the beans from this sample [12].

In the northeast of the state of São Paulo, a temperate region, with moderate rains (average temperature of 20 °C and 30 mm/month of rain), relatively high altitude (800–1000 m), where good quality coffees are usually produced, two samples from the same farm presented above recommended OTA levels. One sample was obtained during storage and contained 10 μg/kg of OTA, while the other sample, derived from the drying patio, contained 48 µg/kg of OTA. Following analysis, authors described that the drying patio was excessively small for the amount of coffee being processed, such that after drying, piles of coffee would remain on the patio. Fog was also considered a problem, since the farm was located in a valley. Finally, the necessary drying time was considered excessive allowing for the growth of fungi and production of OTA [31].

Taniwaki et al. [31] also analyzed farms in western São Paulo, a hot and rainy region (average temperature of 21 °C during growing and harvest and 47 mm/month of precipitation), of relatively low altitude (<800 m), characterized by production of average quality coffee. Authors reported isolating *A. carbonarius* from a visibly moldy sample containing 5 μg/kg of OTA, whose growth was favored by the high temperatures. The investigation showed that the low quality of the samples was due to a broken elevator, which led to humid coffee remaining at the bottom of the compartment where the beans were dried. Another case from the west of the state of Minas Gerais, a temperate and dry region (average temperature of 19 °C and 15 mm/month of precipitation), at a high altitude (>1100 m), producing good quality coffee, was free of OTA producing fungi and OTA, since this farm presented good farming practices throughout all processing stages [12].

According to the World Health Organization (WHO) [38], aflatoxins are toxic substances produced by certain fungi (molds) that are naturally found all over the world and can contaminate food crop plantations and represent a serious threat to human and animal health. *Aspergillus flavus* and *A. parasiticus* are the main fungi responsible for aflatoxin production of public health importance. Under normal favorable conditions in tropical and subtropical regions, including high temperature and humidity, these fungi, which are normally found on dead and decomposing vegetation, may invade food crop plantations. Water stress, damage caused by pests and improper storage may also contribute to the occurrence of molds, in temperate regions as well. Various types of aflatoxins occur in nature, although aflatoxin B1 (AFB1), aflatoxin B2 (AFB2), aflatoxin G1 (AFG1), and aflatoxin G2 (AFG2) are considered the most dangerous for humans and animals because they have been found in all main food cultures. 

Food crops may be contaminated before or after the harvest. Pre-harvest contamination with aflatoxin is mainly limited to corn, cotton, peanut and walnuts. On the other hand, post-harvest contamination can be found in a variety of other cultures such as coffee, rice and spices [38].

Long-term or chronic exposure to aflatoxins present various health consequences. They are potent carcinogens and can affect all organ systems, especially the liver and kidneys. AFB1 is known as a human carcinogen and its ability to cause liver cancer is significantly increased in the presence of hepatitis B virus. It is also an immunosuppressant and reduces resistance to infectious agents such as HIV and tuberculosis [38].

According to Paterson & Lima [39], ideal temperatures for *A. flavus* growth and production of aflatoxin are 33 and 35 °C, respectively, which are superior to ochratoxigenic fungi. High levels of humidity may also favor growth of *A. flavus* and the production of aflatoxin. The minimum water activity for aflatoxin production by *A. flavus* is 0.82, which corresponds to approximately 18.4% humidity. Minimum and maximum temperatures for *A. flavus* growth range from 6 and 10 °C to 25 and 37 °C, respectively, while ideal temperatures for aflatoxin B1 and B2 production range from 16 to 31 °C [40].

Silva et al. [40] analyzed coffee from a farm located 750 to 800 m above sea level in Lavras in the state of Minas Gerais and found the number of isolated fungi in coffee beans, predominantly *A. flavus* and *A. niger*, to have increased during storage. Authors reported that during storage, the number of isolated species in samples stored in jute bags was greater than in samples stored in polystyrene bags, since the latter are less permeable and permit less reabsorption of water than jute bags. 

According to Taniwaki et al. [31], few reports exist regarding *A. flavus* or related species in coffee beans, thus aflatoxin is still not considered a serious problem for coffee.

## 5. Ecophysiology of Toxigenic Fungi under Climate Change

The effect of interactions between environmental factors (temperature × water stress × CO_2_) on the ecophysiology of toxigenic fungi has been recently in focus. Mshelia et al. [41] examined the combined effects of water activity (0.92, 0.95, 0.98 aw), CO_2_ (400, 800, 1200 ppm) and temperature (30, 35 °C and 30, 33 °C for *Fusarium verticillioides* and *F. graminearum*, respectively) on fungal growth and mycotoxin production of acclimated isolates of *F. verticillioides* and *F. graminearum* isolated from maize. They found that elevated temperature and CO_2_ levels applied did not have a significant impact on fungal growth or on mycotoxin production in acclimated *Fusarium* isolates. These findings show the potential of *Fusarium* species to adapt to climate change scenarios. 

Marín et al. [42] studied the effects of ecophysiological factors, temperature and solute potential on growth of *Fusarium verticillioides* and *Fusarium proliferatum* and their regulation of the FUM1 gene, one of 16 genes of the biosynthetic gene cluster responsible for producing fumosnisin, a family of mycotoxins with a significant impact on the quality of maize products. Authors observed that FUM1 gene expression was strongly induced at 20 °C in both isolates under suboptimal growth conditions, despite presenting differences in gene expression patterns with regards to the effect of solute potential. While FUM1 mRNA was induced in response to water stress in *F. verticillioides*, the *F. proliferatum* isolate presented stable expression of the gene under the same conditions, suggesting that there may be different regulatory mechanisms of fumonisin biosynthesis in these species. Furthermore, in environmental conditions that lead to water stress such as droughts, there may be increased risk of fumonisin contamination due to *F. verticillioides*. 

According to Magan et al. [43], episodes of extreme drought, desertification and fluctuations in humid/dry seasons may greatly impact the life cycle of toxigenic fungi. Authors revisited available ecological data regarding optimal and marginal conditions for interaction between temperature and water activity and examined the effects of water stress and a +3 or +5 °C temperature change on growth and mycotoxin production of various toxigenic species. They report that toxigenic fungi normally grow slower and produce similar or lesser amounts of mycotoxins under temperature and water stress, however, in some cases, such as *A. flavus*, they grow and produce aflatoxin B1 (AFB1) at higher temperatures. 

Cairns-Fuller et al. [44] report that environmental factors such as water activity, temperature and CO_2_ concentration play a crucial role in determining growth rates and OTA production by *Penicillium verucosum*. Normally the range of temperature and water activity for mycotoxin production is narrower than for growth, however, *P. verrucosum* was shown to grow and produce OTA within very similar temperatures and water activity ranges. 

Pardo et al. [45] analyzed the growth and OTA production of *A. ochraceus* in green coffee and report that both are influenced by temperature and water activity. Ideal growth conditions for this species are a temperature of 30 °C and water activity of 0.95–0.99, while maximum production of OTA was observed at 20 °C and 0.99 aw. OTA was not produced at 10 °C, regardless of water activity, nor at 0.80 aw. 

Bellí et al. [46] determined the temporal accumulation profile of OTA from *Aspergillus carbonarius* and *Aspergillus niger* isolates in grapes at different water activities, where results show the significant influence of high water activity on OTA production for *A.* section *Nigri* strains. OTA production was shown to be significantly greater for *A. carbonarius* strains than for *A. niger*, however, this was due to one *A. carbonarius* strain (W120) that produced higher amounts than the rest. Water activity of 0.96 combined with a 5 day incubation period resulted in the maximum OTA production, after which the amount of mycotoxin fell over time, reaching a minimum after 20 days of incubation, probably due to degradation by the fungi itself. 

Mitchell et al. [47] investigated the in vitro effect of water activity and temperature on OTA production by *A. carbonarius* isolates and wine grapes, where most strains presented an ideal growth temperature of 30–35 °C, regardless of which solute was used to alter water activity. Growth was not detected at temperatures below 15 °C. Optimum water activity for isolate growth ranged from 0.93 to 0.987, with the largest tolerance to water activity at 25–30 °C. Ideal conditions for OTA production varied by strain where some ideally produced OTA between 15–20 °C and 0.95–0.98 aw. Maximum OTA production after 10 days was between 0.6–0.7 μg/g with an average production of 0.2 μg/g in ideal environmental conditions. 

Medina et al. [48] examined the interaction between water stress, temperature and elevated CO_2_ during growth and analyzed expression of genes involved in aflatoxin biosynthesis (*afl*D and *afl*R) and phenotypic production of aflatoxin B1 (AFB1) by a strain of *Aspergillus flavus*. The study showed that, even though water activity had affected growth, temperature and CO_2_ exposure did not cause a statistically significant effect. At 34 °C, the maximum relative expression of *afl*D occurred under controlled conditions (34 °C, 350 ppm), with a reduction in gene expression under elevated CO_2_ exposure and water stress. A significant increase in *afl*R gene expression was observed at 34 °C, but only at 0.02 aw and 650 ppm of CO_2_. Nonetheless, significant induction of gene expression was observed for *afl*D and *afl*R at 37 °C, 0.95 and 0.92 aw, 650 and 1000 ppm of CO_2_, respectively, suggesting a significant impact in biosynthetic pathways involved in secondary metabolites by the *A. flavus* strain.

## 6. Post-Harvest Microbial Ecology of Coffee Beans

Climate change slowly shapes the balance between hosts, pathogens/pests and the environment [49]. When it comes to toxin producing fungi, a predominance could be swayed from a more suppressive to a more permissive one, or vice versa [50]. As has been seen in Europe, where an increased risk of aflatoxins has been observed in recent years [18], toxigenic fungi could disappear from one environment and appear in others. It has been predicted that over the course of the next century, *A. flavus* may outcompete *A. carbonarius*, with aflatoxins becoming a greater risk than OTA [39], thus the importance of understanding the profound influence of climate change on the biodiversity and ecology of toxigenic fungi. 

Until recently, research has focused on the study of specific microorganisms associated to plants through classic microbiological approaches involving isolation and cultivation. These techniques have been used to study fungal diversity in coffee plantation systems [51,52], which has led to increased understanding of fungal community ecology in this crop. Isolation methods have also been coupled with molecular methods for amplification and first generation (Sanger) sequencing of simple genetic markers of fungi such as 26S and ITS, often after using separation techniques such as denaturing gradient gel electrophoresis [53,54,55].

More recently, next generation sequencing (NGS) technology has led to great advancements in the understanding of the microbial diversity of the environment. In this case, sequences are generated directly from complex microbial communities in environmental samples without the need of isolating and cultivating microbes. Two main approaches are used when probing microbial communities using NGS, namely, high-throughput screening of marker gene amplicons (also referred to as metabarcoding, targeted gene survey or even metagenetics) and shotgun metagenomics. The first involves PCR amplification of highly conserved marker genes, such as those that code for ribosomal subunits 16S/18S and 28S or the internal transcribed spacer (ITS) in fungi. These ubiquitous genes have diverged enough that the polymorphisms in their hyper-variable regions allow taxonomic classification [56]. Following DNA extraction from complex microbial communities, these markers are amplified by the polymerase chain reaction (PCR) and sequenced using NGS. This strategy is often used to decipher the composition and distribution of microbes in an environment and is highly sensitive. 

Shotgun metagenomics, on the other hand, involves direct sequencing of whole genomes present in complex microbial communities in a sample. In this approach, DNA is extracted from samples, sheared and sequenced by NGS, without the need for PCR amplification. Shotgun sequencing can be used to evaluate taxonomic composition and estimate functional potential of microbial communities. Compared to amplicon sequencing, this strategy avoids biases common to amplicon screening [57] and provides better phylogenetic resolution than targeted approaches [58]. Regardless of the approach chosen, NGS results are made up extensive datasets which require elaborate post processing and statistical tools in order to extract information from the data [59].

The diversity of microorganisms plays an important part in the endogenous metabolism of coffee beans, especially at the fermentation stage [60]. At this step, microorganisms are very prevalent, highly variable and difficult to predict [60,61]. While other authors have discussed the microbial diversity associated to other components of the coffee plant, such as the rhizosphere, the episphere and endosphere [62,63,64,65,66,67,68], we have focused our review on the postharvest microbiota which, as described above, has been shown to be highly relevant to coffee quality and mycotoxin production. 

Up until the present, ten works have been published using next generation sequencing approaches to evaluate the impact of post-harvesting methods on coffee bean microbial community profiles (Table 2). These studies were all conducted with arabica coffee, yet they were dispersed across 3 continents including three studies in Ecuador [69,70,71], one in Brazil [72], one in Honduras [73], one in Mexico [74], one in Colombia [75], two in Australia [76,77] and one in China [60]. Of the 10, only two studies depended on shotgun metagenomic sequencing and for the targeted amplicon studies authors chose different variable regions of the 16S rRNA gene, and either ITS or 18S to probe fungi.

For a detailed list of the relative abundances of the microorganisms identified by 7 of the 10 studies mentioned herein, we refer the reader to Duong et al. [66] who has already provided a descriptive survey of the microorganisms identified through these studies (along with previous studies). Instead, we prefer to highlight some of the main hypotheses stemming from these works and discuss the future perspectives for better understanding of the influence of environmental factors on different stages of microbial life cycles. 

Despite differences in study designs, some of the works listed collected samples during the entire post-harvesting chain and noted a clear difference between the microbial community structures of freshly harvested cherries when compared to samples further downstream in coffee processing stages. De Bruyn et al. [69], for example, found bacteria pertaining to *Enterobacteriaceae* (*Klebsiella pneumoniae*), acetic acid production (*Gluconobacter* spp.) and soil (*Dyella kyungheensis*) and a very small proportion of lactic acid bacteria (*Leuconostoc mesenteroides*/*pseudomesenteroides*) in freshly harvested cherries. In regard to fungal diversity, *Pichia kluyveri* was highly abundant. Similar contamination of freshly harvested cherries including *Acetobacter*, *Gluconobacter*, *Leuconostoc pseudomesenteroides* and *Pichia kluyveri*, among others, were observed on the same farm by Zhang et al. [70]. De Carvalho Neto et al. [72] discussed that possible habitat origins of these initial groups are human contact (*Pseudomonas* sp., *Enterobacter*,) soil or aerial parts of coffee plants (*Mesorhizobium*), the water source used (*Planctomyces*) and the air (*Janthinobacterium*). More recently, Da Silva Vale [73] used NGS to investigate the role of farm microbiota (including tree leaves, surface of tree cherry, soil, ground leaves, surface of ground fruit, water sources, surface of over ripe fruit, depulped fruit and fermentation) and determined that coffee fruits are themselves are the most probable origins of the beneficial microorganisms for the fermentation process. Other sources such as leaves, fruit surfaces and soil may transfer unwanted microorganisms to coffee beans and should be avoided. 

Eight of the ten studies went on to evaluate microbial community structure during wet fermentation over time [60,69,71,72,75,76,77,78]. All of these works demonstrated strong adaptation of lactic acid bacteria to the coffee fermentation environment. Lactic acid bacteria, predominantly *Leuconostoc*, asserted a quantitative prevalence over other groups such as *Enterobactereae* and acetic acid bacteria as fermentation progressed. Nonetheless, longer fermentation times (>24 h) resulted in a microbial shift from leuconostocs to acid-tolerant lactobacilli [69,76,78]. In general, lactic acid bacteria have been shown to contribute to inhibition of pathogens, spoilage microbes and toxin producing fungi [75].

In wet fermentation, most studies found limited fungal diversity, with *Pichia governing* the process. The *Pichia* species identified included *P. kluyveri* [69,78], *P. nakasei* [75] and *P. kudriavzevii* [71,76,77]. Both common and region-specific species have been found during wet fermentation of coffee beans [76]. Yeasts are well known for their production of secondary flavor metabolites such as organic acids, esters and aldehydes. The interaction of these yeasts with the dominant lactic acid bacteria present during the fermentation process provide a complex association which have been shown to promote desired sensory attributes in other fermented foods such as wine, sourdough and yogurt [75]. Further studies are necessary to shed light on the interaction between lactic acid bacteria and yeast in the coffee fermentation process. 

De Bruyn et al. [69] went beyond the microbial community structure of wet fermentation and compared it to dry fermentation as well. A clear distinction was observed between the wet and dry processing of coffee, with a higher prevalence of acetic acid bacteria *Acetobacter* and *Gluconobacter* and greater fungal diversity. This was also confirmed by increased metabolite concentrations of acetic acid, ethanol, glycerol and mannitol. Authors suggest that the microorganism profile of dry-processed coffee beans may imply a slow but observable migration of microbial metabolites to the endosperm, resulting in higher bitterness and astringency levels than wet processed ones. 

Zhang et al. [60] also analyzed the microbial community dynamics during the entire post-harvest process and compared a depulping (DP) to a demucilaging (DM) step prior to fermentation. The main difference between these two steps is the amount of mucilage that remains on the bean during fermentation, in which the DM treated beans have most of the mucilage scraped off prior to fermentation. Processing type (DP or DM) was shown to account for almost 30% of the variation in the microbial community composition in this work. The absence of the mucilage during fermentation led to a shift in preference from *Leuconostoc* to *Lactococcus* in the DM treated beans. This shift reflects a preference for substrate concentrations, giving certain microbial communities a competitive advantage to increase their relative abundances. 

Both De Carvalho Neto [72] and De Oliveria Junqueira [75] pointed out that adoption of culture-independent methods greatly increasing the capacity of identifying microbial diversity. While nine bacterial genera had been reported in previous studies using culture-dependent methods, De Carvalho Neto [72] identified over 80 genera using a targeted amplicon approach. Just a year later, De Oliveira Junqueira [75] identified over 157 genera using a similar approach. Both Zhang et al. [60,78] and Pothakos et al. [71] were the only ones to apply a shotgun metagenomics approach to the coffee post harvesting process and showed the strength of the technique to identify genes, predict functions and build networks, besides taxonomic classification of microorganisms. While only true metatranscriptomics can elucidate actual gene expression, the functional prediction based on genes has underlined the contributions of the different microbial groups to wet coffee processing. 

The effect of climate change on microbial diversity and the resilience of toxigenic fungi through culture-independent technology has yet to be explored in the coffee crop. Drought is the main environmental restriction affecting coffee growth and production [79], not only for arabica coffee but also for robusta, which until recently was considered resistant to temperature increases, having been recently demystified by research conducted in Southeast Asia [80]. Some studies provided evidence that potential mycotoxigenic fungi may not be affected by the CO_2_ treatments [81], nonetheless, these studies need to be better designed in order to include other climatic factors linked to the natural microbiome associated with coffee production/productivity [82]. Thus, it is still mandatory to design indirect means of assessing the vulnerability of coffee microbiota to temperature change in the field allied to stress and other climate factors, such as water and CO_2_, to verify how these climatic factors influence the population fluctuation of microorganisms and microbial succession directly associated with the production of coffee beans. This information is still necessary for decision-making related to the sustainability of the coffee sector, since its production is considered the main agricultural product in some Brazilian states and may be jeopardized by climate change [79].

These studies can elucidate the elements underlying the plasticity and vulnerability of coffee under the future conditions which becomes a fundamental basis for plant breeders to obtain new/more adapted genotypes [83,84,85] as strategies for maintaining the safety of the coffee production chain in the fields in the near future.

## 7. Multi-Omics to Study the Coffee Microbiome in a Climate Change Scenario

Previously, we described meta-genetics (targeted amplicon sequencing) and metagenomics (shotgun sequencing) as next generation sequencing approaches to study diversity and structure of complex microbial communities. However, these are just a few within an arsenal of high-throughput techniques that have contributed to a rapid expansion of data and facilitated a significant increase in our knowledge of biological and biochemical processes. The techniques of transcriptomics, proteomics and metabolomics, together named multi-omics approaches, are already being used to gain an understanding of the functional capabilities of isolated microorganisms [86,87]. Fortunately, recent studies have expanded beyond identifying a few microorganisms to characterizing more and more complex microbial communities and their impact on the plant host. While multi-omics approaches are still crucial to clarify yet unanswered questions directed at single microorganisms and the interaction with their hosts, technology has allowed us to probe the functions of entire microbial communities using functional meta-omics approaches, such as meta-transcriptomics, meta-proteomics and meta- or community metabolomics [70]. The integration of these tools may enhance our functional understanding of the coffee microbiome, including how it responds to changes, how its members interact, and how it impacts safety and quality of the final product. 

Metabolomics is the screening of multiple low molecular weight metabolites which provide the closest insight into the physiology of the cell under different environmental, genetic, pathological or developmental conditions [86]. The field involves a range of chromatographic techniques coupled with mass spectrometry (MS) or nuclear magnetic resonance (NMR). Some of these include gas chromatography coupled to MS (GS-MS) and tandem MS (GC-MS/MS), liquid chromatography coupled to MS (LC-MS) and to tandem MS (LC-MS/MS) and liquid chromatography coupled to NMR and MS (LC-NMR-MS). Aditiawati et al. [88] used a metabolic profiling approach to evaluate controlled fermentation of arabica coffee beans in Indonesia. Comparative GC-MS analysis was able to show that coffee bean fermentation with bacterial isolates from civet feces resulted in alteration of metabolite profiles when compared to control, while still maintaining the characteristics of coffee from three different origins (Sumedang, Aceh and Bali). Also, shorter fermentation (4 h) resulted in increased sugars while longer fermentation (8 h) led to more amino acids which also affected the flavor characteristics of these two conditions. Several of the works mentioned in the previous chapter that probed the microbial profile during coffee post-harvesting and fermentation also investigated the metabolic profile of the coffee bean during these processes [69,75,76,77,78]. Of note, was the strategy used by Elhalis et al. [77] in which the role of yeasts in fermentation was investigated by suppression of this group by Natamycin, a food-grade anti-fungal agent. Suppression of yeasts by Natamycin had a significant impact on the production of key microbial metabolites during coffee fermentation, such as glycerol, alcohols, esters, aldehydes and organic acids. The growth of yeasts was also important in inhibiting filamentous fungi and undesirable metabolites, such as acetic acids.

Transcriptomics is the study of gene expression by measuring the complete set of RNA transcripts within a cell, which varies under different times or conditions. Currently, both sequencing-based and hybridization-based methods exist for examining the transcriptome, however, RNA-Seq, which is based on next generation sequencing has considerable advantages for examining the transcriptome profile structure, such as the detection of novel transcripts and identification of splice variants. Few studies have used transcriptomics to investigate the interaction between coffee and microorganisms. Florez et al. [89] used RNA-Seq to evaluate the response of two coffee genotypes (Caturra, resistant and Hibrido de Timor, susceptible) to *Hemileia vastatrix*, the fungal agent which causes coffee rust disease, and identified genes and biological pathways that are involved in resistance to the pathogen. 

Other authors have focused on the transcriptome of toxigenic fungi while probing abiotic factors affecting mycotoxin production as well as the fungus-plant crosstalk, mostly with maize [67,90,91,92,93]. While examining *A. flavus* activity, both Zhang et al [90] and Yu et al. [91] observed a strong transcriptome response when comparing different water activities and temperatures, respectively. A total of 5362 genes were shown to be differentially expressed between treatments with an aw of 0.99 and 0.93 [90], while 1153 genes were differentially expressed between temperatures of 30 and 37 °C [91]. Gilbert et al. [94] went on to demonstrate that CO_2_ levels had a measurable impact on the fungal transcriptome. Changes in temperature and water availability at usual CO_2_ (350 ppm) levels regulate gene expression differently than when these conditions were altered at higher CO_2_ levels (1000 ppm). Taken together, these results indicate that the three abiotic factors associated to climate change have a measurable impact on molecular events in fungi. 

When looking at entire complex communities of microorganisms, meta-transcriptomics is emerging as an important complement to metagenomic studies, since their combination not only improves microbial genome assembly and gene prediction, but also enables identification of expressed genes under specific conditions. However, metatranscriptomics studies of the full microbial communities associated with plants are very limited. A recent example of a combined metagenomic, metatranscriptomic and metabolomic approach was carried out in the work of Verce et al. [95] in which the functions of the microbial community present during the fermentation of cocoa was evaluated over time. This strategy provided a deeper characterization of the metabolic activities of previously established key players as well as insight into previously overlooked microbes, microbial processes and interactions within them [95].

Since proteins are important components of biochemical pathways, identifying proteins is essential to revealing molecular mechanisms underlying biological processes. Similar to the transcriptome, the proteome, or the complete set of proteins in an organism, is dynamic and varies due to both biotic and abiotic factors [86]. The study of proteins has become quite complex because of the variability in the number of protein species per gene due to alternative splicing, post-translational modifications and especially interactions, considering that protein complexes, rather than individual proteins are responsible for biochemical processes [96]. Using high resolution mass spectrometry, proteomics was used to identify proteins produced by a high and a low OTA producing *A. carbonarius* [97]. Nine differentially expressed proteins were identified and possible functional roles were speculated contributing to a better understanding of OTA production. Despite powerful proteomic tools, only a small fraction of the cell proteome and that of a few organisms has been characterized so far. Just as with the transcriptome, it is also possible to investigate the metaproteome of all the organisms present in complex microbial communities, regardless of their phylogenetic origin, which can lead to greater understanding of host-microbiome interactions. 

Despite being extremely challenging and still in their early phases, integration of multi-omics and multi-meta-omics data promises to comprehensively characterize microbiome composition and function as well as their metabolites. As the throughput of these tools increase and costs decrease, they will become common analytical methods for microbiome-based studies. This will be particularly important in understanding the impact that climate change may have on coffee, microbes and the production of mycotoxins. 

## 8. Concluding Remarks

Over the last decade, an apocalyptic scenario has been foreseen with projections of extensive reduction of area destined to coffee production [24,83,98]. According to De Sousa et al. [26], in the most positive scenario, coffee could be replaced by cocoa; however, in addition to the reduction of arable land, this transition could pose additional risks to coffee farming. Still in this scenario, other fungi and mycotoxins, such as aflatoxins and aflatoxin-producing fungi, may become more competitive, becoming more prevalent in coffee than OTA, while being more toxic, promote a greater risk to human health.

Thus, a closer look by authorities such as FAO, especially the Ministries of Agriculture and Food from coffee producing countries, is essential to stimulate the gradual replacement of suitable coffee production areas, in accordance with climatic zoning, to avoid the shortage of coffee in the world market.

Since consumption projections have also increased considerably all around the world, it becomes mandatory that the authorities and humanitarian aid organizations come together to support, protect, and lead the necessary fast decision-making process. This is essential to adopt solutions (agroforestry and agronomic mitigations, new cultivars adapted to high temperatures) for the maintenance of local agricultural economies, the maintenance of families on the countryside and to avoid shortages in the international coffee market.

## Figures and Tables

**Table 1 microorganisms-09-00858-t001:** Coffee production in a global warming scenario.

Type of Study	Location	Negative Scenario	Mitigation Strategy	References
Review article	Brazil	Strong decrease in coffee production and productivity in Brazil.	The coffee crop will tend to move south and to uphill regions.	[22]
Review article	Worldwide	Coffee supply chains will be affected by significant disruption; coffee production will decrease globally; Increase in the price of coffee.	Actions to reduce greenhouse gas emissions are mandatory.	[23]
Review article	Worldwide	Coffee plant’s physiological performance at elevated atmospheric carbon dioxide (CO_2_) concentration	Suitability of coffee may be lower than previously assumed. Priorities for further research to improve understanding on how the coffee plant will respond to present and progressive climate change.	[24]
Analysis of climate data. Modeling and validation of climate suitability.	Nicaragua	Sensitivity of *Coffea arabica* and the likely impact of climate change on coffee suitability, yield, increased pest and disease pressure and farmers’ livelihoods.	Lower altitudes, whereas the same areas may undergo transformative adaptation in the long term. At higher elevations incremental adaptation may be needed in the long term.	[25]
Integrating trees in combined agroforestry systems to ameliorate abiotic stress.	Mesoamerica	Significant reductions in coffee and cocoa agroforestry production areas.	Transforming agroforestry systems by changing tree species composition may be the best approach to adapt most of the coffee and cocoa production areas. Urgency for land use planning considering climate change effects and to assess new combinations of agroforestry species in coffee and cocoa plantations.	[26]

**Table 2 microorganisms-09-00858-t002:** Next generation sequencing approaches to studying the effect of post-harvesting on coffee microbial dynamics.

Coffee Species	Location	Study Design	NGS Strategy	References
*Coffea arabica* L. var. typica	Nanegal, Ecuador	Evaluation of two different wet and dry post-harvest methods on microbial community structure and metabolite profiles over a 15 and 28 day time period, respectively.	Targeted Amplicon Sequencing; Illumina MiSeq sequencing of the V4 region of 16S rRNA (bacteria) and ITS1 region (fungi).	[69]
*Coffea arabica* L.	Veracruz, Mexico	Evaluation of storage of green coffee beans in jute bags for one year with sampling once a month.	Targeted Amplicon Sequencing; Illumina MiSeq sequencing of the V4 variable region of 18S rRNA gene (fungi).	[74]
*Coffea arabica* var. Catuaí	Cerrado Mineiro, Minas Gerais, Brazil	Evaluation of bacterial community composition at 0, 12 and 24 h of fermentation.	Targeted Amplicon Sequencing; Illumina MiSeq sequencing of the V3 region of 16S rRNA (bacteria only).	[72]
*Coffea arabica* L.	Buesaco, Colombia	Evaluation of microbial communities in liquid fraction of "washed" fermenting coffee bean at 0, 6, 12, 18, 24, 36 and 48 h.	Targeted Amplicon Sequencing; Illumina based sequencing of V4 region both of 16S and 18S rRNA genes (bacteria and fungi, respectively).	[75]
*Coffea arabica* L. var. Typica	Nanegal, Ecuador	Evaluation of microbial community profile, metabolites and bean chemistry during the entire wet processing chain and evaluated sensory quality of final coffee product.	Targeted Amplicon Sequencing; Illumina MiSeq sequencing of the V4 region of 16S rRNA gene (bacteria) and ITS1 region of the 26S gene (fungi).	[78]
*Coffea arabica* var. Catimor	Yunnan, China	Compared effect of demucilaging and depulping, fermentation duration and soaking on the microbial community composition and meta-metabolomic profiles.	Targeted Amplicon Sequencing (see Zhang et al., 2019a) AND Shotgun Metagenomics	[60]
*Coffea arabica* var. Bourbon	Teven, Australia	Evaluation of microbial composition during wet fermentation over time (36 h).	Targeted Amplicon Sequencing; Illumina MiSeq sequencing of the V3-V4 region of 16S rRNA gene (bacteria) and ITS region of the 26S gene (fungi).	[76]
*Coffea arabica* L. var. Typica	Nanegal, Ecuador	Evaluation of microbial dynamics during wet fermentation comparing standard (16 h) and extended (64 h) protocols.	Shotgun metagenomics, Illumina MiSeq	[71]
*Coffea arabica* var. Bourbon	Teven, Australia	Evaluation of role of yeasts during wet fermentation by adding Natamycin, a food-grade anti-fungal agent.	Targeted Amplicon Sequencing; Illumina MiSeq sequencing of the V3-V4 region of 16S rRNA gene (bacteria) and ITS region of the 26S gene (fungi).	[77]
*Coffea* sp.	Teupasenti, Honduras	Evaluation of coffee farm microbiome and contribution to fermentation	Targeted Amplicon Sequencing; Illumina sequencing of 16S (Bacteria) and 18S rRNA gene (Fungi)	[73]

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
