# Peer review of "Brazilian Coffee Production and the Future Microbiome and Mycotoxin Profile Considering the Climate Change Scenario"

_microorganisms, 2021, doi:10.3390/microorganisms9040858_

Round 1

Reviewer 1 Report

The prepared work concerns the very interesting impact of climate change and microclimate on the microbiome inhabiting coffee plantations and bushes in Brazil. Contamination with mycotoxins of this raw material has been a very serious problem for years. It seems that forecasts of progressive climate change and hence changes in the microbiome may have serious consequences for consumers. Mycotoxins as a food contaminant pose a huge challenge for producers.

Please answer the following question or include additional issues in the article.

Question 1. Are trends of this type observed in other raw materials or in other countries? Perhaps the article should be supplemented with this issue.

Author Response

Firstly, we would like to thank you for your good comments, observations and considerations.

 Regarding trends in the impact of climate change and the microclimate on the microbiome in other raw materials or in other countries  this question was answered in the text and can be found on pages 3 and 4. 

"Battilani et al.[18] reported that changes in mycotoxin occurrence patterns, such as aflatoxins in crops in Europe due to climate change, are a cause for concern. Those authors pointed that official control measures contribute to the global effort to reduce the risks of exposure to aflatoxins through food and feed intake by humans and animals, however, specific action plans need to be directed to the production chain. In addition to these measures, other recommendations have been suggested to minimize the impacts of global climate change as i) the use of the modeling approach as a support tool to strengthen the management of aflatoxin to prevent human and animal exposure, ii) the adoption of new strategies supported by forecasts, iii)  biological control through with use of  atoxigenic strains , capable of displacing the toxigenic populations of the fungus iv)use of risk maps as a communication tool for stakeholders, especially for farmers and livestock keepers, v) management tool to highlight areas at risk of mycotoxins contamination, in order to prioritize their control and intervention strategies [19–21]."

For  coffee a new paragraph was added: 

"The impacts of climate change on the coffee microbiome have already been reported in East Africa, where the coffee borer (Hypothenemus hampei) has benefited from the increase in temperature causing major damage to coffee crops and expansion in its distribution area. The situation with H. hampei is expected to worsen in the current production areas of C. arabica in Ethiopia, Kenya, Rwanda and Burundi". 

The Table 1 also  brings others exemples  worldwide as well as on   Guatemala, Nicaragua and Mesoamerica.

Reviewer 2 Report

The manuscript focuses on how Brazilian coffee trees and crops may be affected by climate change increasing accumulation of mycotoxins and inducing changes in the microbiomes. The review is well-organized, well-referenced and would be of interest to other readers. I would support acceptance after minor revisions to mostly fix grammatical errors. Some examples include:

use of uppercase for common nouns

p 2- species name should be lowercase (C. arabica)

p 2, line 2- ‘produce’ should be ‘producer’

p 3- ‘Porto Rico’ should be ‘Puerto Rico’

p 5- ‘micotoxins’ should be ‘mycotoxins’

A short section on potential mitigation strategies would be useful. If they haven’t already, the authors may consider including any of the following references relating to the effects of temperature or climate change on mycotoxin accumulation in the attachment.

Author Response

Authors thanks to  the reviewer for  the positive comments.

-The grammar and misspelled words were  corrected.   There was also a revision of English throughout the manuscript.

- According your suggestion it was  included a new section with data  about "Potential mitigation strategies" . This new section was added in the manuscript  and some paragraphs in the text that contained this subject (mitigation strategies) including the Table 1  have migrated to this new section.

Some information data (highlighted in red)  was included on pages 4 and 5,  where Battilani and colleagues suggest using the modeling approach as a support tool to strengthen the management of aflatoxin in preventing exposure human and animal, as well as the adoption of new strategies based on predictions, such as biological control through strains of A. flavus atoxigenic, capable of displacing the toxigenic populations of the fungus, widely applied in risk areas in the USA and Africa.

-We also added to the text some new references indicated by the reviewer and we are grateful for the suggestion.

We appreciate the considerations.